# Diabesity in Elderly Cardiovascular Disease Patients: Mechanisms and Regulators

**DOI:** 10.3390/ijms23147886

**Published:** 2022-07-17

**Authors:** David García-Vega, José Ramón González-Juanatey, Sonia Eiras

**Affiliations:** 1Cardiology and Intensive Cardiac Care Department, University Hospital, 15706 Santiago de Compostela, Spain; dav.garcia.vega@gmail.com; 2Cardiology Group, Health Research Institute, 15706 Santiago de Compostela, Spain; 3CIBERCV, 28029 Madrid, Spain; 4Translational Cardiology Group (Laboratory 6), Health Research Institute, 15706 Santiago de Compostela, Spain

**Keywords:** diabesity, cardiovascular disease, epicardial fat

## Abstract

Cardiovascular disease (CVD) is the leading cause of death in the world. In 2019, 550 million people were suffering from CVD and 18 million of them died as a result. Most of them had associated risk factors such as high fasting glucose, which caused 134 million deaths, and obesity, which accounted for 5.02 million deaths. Diabesity, a combination of type 2 diabetes and obesity, contributes to cardiac, metabolic, inflammation and neurohumoral changes that determine cardiac dysfunction (diabesity-related cardiomyopathy). Epicardial adipose tissue (EAT) is distributed around the myocardium, promoting myocardial inflammation and fibrosis, and is associated with an increased risk of heart failure, particularly with preserved systolic function, atrial fibrillation and coronary atherosclerosis. In fact, several hypoglycaemic drugs have demonstrated a volume reduction of EAT and effects on its metabolic and inflammation profile. However, it is necessary to improve knowledge of the diabesity pathophysiologic mechanisms involved in the development and progression of cardiovascular diseases for comprehensive patient management including drugs to optimize glucometabolic control. This review presents the mechanisms of diabesity associated with cardiovascular disease and their therapeutic implications.

## 1. Diabesity in Elderly Patients

Lifestyle changes related to an increase in caloric intake, reduced physical activity and an increase in life expectancy have developed an ageing population with obesity, diabetes and their comorbidities. It has contributed to an exponential increment of healthcare costs during the last decades. In 2020, 650 million people were living with obesity and 460 million with diabetes [1] (Figure 1). Some registers have defined differential diabetes-associated risk factors with respect to sex. While education and ageing were diabetes-risk factors in male, physical inactivity and obesity were risk factors in female. The prevalence of diabetes varies in different age groups, posing a higher risk to the old compared to the young population [2]. The main causes of this accelerated diabetes prevalence in ageing societies are similar to those in younger people, which include unhealthy obesogenic diets and reduced physical activity [3]. Since incidence of diabetes increases with ageing, diabetic older adults represent the largest population of diabetic subjects, this group being particularly vulnerable to cardiovascular disease (CVD). The relationship between CVD, diabetes and ageing is explained, in part, by the negative impact of these conditions on vascular function [4]. Given the current increase in life expectancy, there is also a higher prevalence of obesity. Both conditions are the leading cause of health problems, disease risk and death. Ageing increases abdominal obesity, independently of body weight, sex or race, which is one of the major contributors to insulin resistance and metabolic syndrome [5,6,7]. The visceral fat accumulation is strongly associated with ectopic fat deposition in skeletal muscle, heart, liver, pancreas or blood vessels, a trend leading to lipotoxicity [8,9,10] and an increase in pro-inflammatory cytokines [11].

## 2. Diabesity and Cardiovascular Disease

Cardiovascular disease (CVD) is the leading cause of death in the world. In 2019, 550 million people were suffering from CVD and 18 million of them died as a result (Figure 1). Most of these people had associated risk factors such as high fasting glucose, which caused 134 million deaths, and obesity, which caused 5.02 million deaths [12]. In addition, 75% of type 2 diabetes mellitus (T2DM) patients die as a consequence of CVD, including coronary artery disease (CAD) [13]. Their risk of myocardial infarction is comparable to those nondiabetic patients with previous myocardial infarction. Their 5-year mortality rate is twice that of nondiabetic subjects. In addition, the risk of CAD increases in diabetic patients by 11% for each 1% increment in haemoglobin A1c (HbA1c) greater than 6.5% [14]. The Action in Diabetes and Vascular Disease: Preterax and Diamicron MR Controlled Evaluation (ADVANCE) trial showed that every 5-year increase in T2DM duration enhances age-adjusted macrovascular events by 49%, including cardiovascular death, nonfatal myocardial infarction and nonfatal stroke [15]. T2DM is often directly linked to obesity, with approximately 80% of those with T2DM being overweight or obese [16]. In these individuals, cardiovascular mortality increases by 40% for every 5-unit increase in body mass index (BMI) above 25 [17]. Median survival in patients with BMI between 30 and 35 kg/m^2^ or between 40 and 45 kg/m^2^ is reduced by 2–4 years or 8–10 years, respectively [18]. The CARDIA study included patients with several obesity phenotypes (with healthy and unhealthy metabolism). Its analysis has demonstrated the highest risk for coronary artery calcification (CAC) progression in obese subjects with unhealthy metabolism and the lowest risk in non-obese subjects [19]. Ageing is also considered the major risk factor of CVD. In fact, the incidence and severity of subclinical and clinical manifestations of CVD steeply increase with age [20,21], even in those without risk factors [22]. Heart failure (HF) is another major cause of morbidity and mortality of CVD. The incidence of HF hospitalization (adjusted for age and sex) are two times higher in patients with diabetes [23,24,25]. Its prevalence increases with age: from around 1% to >10% in those aged ≥70 years. In addition, prediabetes and diabetes are highly represented among patients with HF (25–40%), with relevant effects on their prognosis [26]. The Danish Investigations of Arrhythmia and Mortality ON Dofetilide (DIAMOND) study suggested that 16% of diabetes and 50% of HF preserved ejection fraction (HFpEF) showed the influence of diabetes (sex-independent) on death risk in hospitalized patients with congestive HF (31% of 1-year mortality and 50% of 3-years mortality) [17]. The Candesartan in Heart Failure Assessment of Reduction in Mortality and Morbidity (CHARM) trial also demonstrated a higher risk of hospitalization for HF and/or death in a sex-independent manner [27]. Elderly patients with uncontrolled diabetes have higher risk of HF progression, as observed in a study on atherosclerosis risk in communities (ARIC) [28]. Another important CVD associated with diabetes is atrial fibrillation (AF). This is the most common sustained cardiac arrhythmia, and its prevalence is expected to double over the next three decades [29]. One of the reasons might be explained by increased survival [30]. The data in the ORBIT-AF register show an association between diabetes and AF development, symptoms burden and lower quality of life, and between increased hospitalization and mortality rates [31]. The ADVANCE study also demonstrated that T2DM patients with AF had higher risk of major coronary events, stroke, HF, cardiovascular death and all-cause mortality compared to those without AF [32]. Obesity and epicardial adiposity can enhance the AF risk [33,34] and reduce the ablation efficiency [35]. Some of the described mechanisms of obesity associated with CVD are related to hemodynamic alterations that make the subjects predisposed to changes in cardiac morphology and ventricular dysfunction and hypertrophy, neurohormonal and metabolic abnormalities, such as increased sympathetic nervous system tone, activation of the renin–angiotensin–aldosterone system, insulin resistance with hyperinsulinemia, leptin resistance with hyperleptinemia, adiponectin deficiency, lipotoxicity and lipoapoptosis. They are described in more detail in the following sections.

## 3. Physiopathological Mechanisms

### 3.1. Cardiac Structural Changes

#### 3.1.1. Coronaries-CAD

The coronary circulation provides oxygen and substrates to the myocardium to ensure its normal function and viability. Owing to the limited anaerobic capacity of the heart, coronary vascular resistance is continuously regulated to deliver sufficient quantities of oxygen to meet any change in the demand of surrounding myocardial tissue. Regulation of coronary blood flow is understood to be dictated through multiple mechanisms including extravascular compressive forces (tissue pressure), coronary perfusion pressure, myogenic, local metabolic and endothelial, neural and hormonal influences. Blood is delivered via epicardial coronary arteries that further divide and penetrate into the myocardium. Coronary arteries that exceed ~0.5 mm thickness receive blood supply from the vasa vasorum externa, which is a specialized microvasculature within the adventitia. Recent data support a link between the expansion of the vasa vasorum with neointimal formation and atherosclerotic disease progression [36]. Coronary artery caliber is reduced in women and even more in diabetic patients [37], whose collateral vessel development is also impaired. The endothelial cells of coronary arteries regulate vascular function and structure. In physiological conditions, their synthesized and released active substances maintain vascular homeostasis, ensure adequate blood flow and nutrient delivery, and prevent thrombosis and leukocyte diapedesis [38]. Vascular ageing is associated with structural changes in endothelial and smooth muscle cells and extracellular matrix of vessel walls [39]. These physiopathological mechanisms increase intima and media thickness, stiffness and dilatation of central elastic arteries, affecting their ability to expand and contract in response to pressure changes. Diabetes also affects vascular function and calcification in young patients that contributes to early coronary atherosclerotic changes [40]. In particular, diabetes affects the prognosis of young diabetic patients, especially females, after myocardial infarction [41]. There is an arterial stiffness and an increase in pulse wave velocity [42], which worsen with ageing [43]. The presence of T2DM in CAD is an important factor associated with the election of revascularization strategy [44] because it is a calcified and diffuse multivessel disorder [45]. Moreover, the CORDIOPREV study has demonstrated a greater reduction of cholesterol efflux capacity and peripheral artery disease in diabetic and coronary heart disease patients [46]. The Look AHEAD study showed that metabolic dyslipidaemia is an additional factor for CVD events in these patients [47]. Epicardial fat might participate in this lipidomic disorder since its profile is highly changed in patients with diabetes and CAD [48]. Visceral adiposity increases inflammatory cytokines [49] and prothrombotic state [50] and inflammation promotes oxidation of low-density lipoprotein (LDL) [51] that accelerates the atherosclerosis process [52]. As abdominal visceral adiposity increases, there is a higher risk of noncalcified plaques [53]. A PARADIGM sub-study has demonstrated that plaque progression is also dependent on ageing [54].

#### 3.1.2. Myocardium-HF

The study on Normal Reference Ranges for Echocardiography (NORRE) is the first European large multicenter study of accredited echocardiography laboratories of the European Association of Cardiovascular Imaging (EACVI) [55] to establish the parameters in normal subjects. The left ventricular and atrial mass, dimensions and volumes differ between men and women. While they are higher in men, the left ventricular ejection fraction (LVEF) is higher in women. However, in both sexes, there is a decrease with ageing [55]. The Dallas Heart Study, which included subjects without CVD and seven years of follow-up, has demonstrated the association between weight gain and increase of left ventricular mass, wall thickness and concentricity [56]. It has been confirmed by other cross-sectional studies [57]. Cardiac hypertrophy is caused by the heart’s adaptation to hypertension [58]. However, longer exposure causes molecular changes that lead to HF [59]. The main cardiac structural changes described in obese animal models with high fat diet are the ventricular septum diastolic and systolic thickness. Moreover, they have a disordered myocardial structure and loss layer arrangement [60]. The main consequences are (a) decreased left ventricular filling capacity (b) increased chamber stiffness and impaired relaxation and longer isovolumetric relaxation times (c) reduction in early-diastolic-filling E-wave-to-atrial-contraction-late-filling (A-wave) ratio, longer deceleration times, higher E-wave-to-early-diastolic-mitral-annular-velocity (e’) ratio, and impaired left ventricular (LV) compliance [61]. A right ventricle remodeling was also observed in subjects with class III obesity (BMI > 40 kg/m^2^). The right ventricle end-diastolic and systolic volumes were increased by 17% and 26%, respectively. The eccentric and concentric remodeling was also reflected in a 12% left ventricular end-diastolic volume and 24% left ventricular mass increment [62]. Although some subjects had a healthy metabolic profile, they presented changes in left ventricular geometry [63]. A UK Biobank Cardiovascular Magnetic Resonance sub-study was able to demonstrate early changes in cardiac morphology and function in patients with diabetes mellitus. They detected reduced right and left chambers without changes in ejection fraction [64]. Some authors have described four possible stages in diabetic cardiomyopathy. The first one is the asymptomatic initial phase, based on left ventricular hypertrophy with preserved ejection fraction, the second one is the dilatation stage with reduced ejection fraction, the third one is the stage with systolic and diastolic dysfunction, micro-angiopathy, hypertension and myocarditis, and the final stage concerns refractory HF with ischemia, infraction and remodeling [65]. The diabetic patient’s microcirculatory dysfunction usually precedes structural myocardial changes. They have a lower coronary flow reserve (CFR) and left atrial reservoir strain, which is an early marker of diastolic dysfunction [66]. The increased BMI exacerbates the reduction of left atrial strain and strain rate in T2DM without changes in left atrial volumes. Moreover, the left atrial reservoir and pump strain were associated with left ventricular global longitudinal strain (GLS), global circumferential strain (GCS) and left atrial conduit with left ventricular peak diastolic strain rate (PDSR). These results suggest atrioventricular interaction with cardiac changes of diabetes [67].

#### 3.1.3. Nervous System (Cardiac Autonomic Neuropathy)-AF

The heart rhythm is controlled by the autonomic nervous system (ANS), which comprises sympathetic and parasympathetic fibers. Noradrenaline sympathetic innervation is heterogeneous and less dense in the cardiac apex. It enhances the contractility of cardiomyocytes through β1 receptors, coupled to the Gs protein; intracellular cAMP levels increase, leading to protein kinase A activation and phosphorylation of L-type calcium channels, delayed rectifier potassium channels, phospholamban and type 2 ryanodine receptors that augment heart rate. However, the released acetylcholine by postganglionary axon terminals of the parasympathetic fibers through M2 muscarinic receptors and potassium channels decreases heart rate and contractility [68]. ANS and endocrine systems are controlled by the hypothalamus, which regulates cardiac output [69]. ANS dysfunction with a depressed parasympathetic tone and an increased sympathetic activity was described in insulin resistance [70], obesity [71,72] or ageing overall in postmenopausal women [73]. Cardiac Autonomic Neuropathy (CAN) is a diabetic neuropathy associated with mortality independently of other cardiovascular risk factors [74]. In its early stages, CAN may be completely asymptomatic and detected only by decreased heart rate variability with deep breathing. Advanced disease may be associated with resting tachycardia (>100 bpm) and orthostatic hypotension (a fall in systolic or diastolic blood pressure (BP) by >20 mmHg or >10 mmHg, respectively, upon standing without an appropriate increase in heart rate). Mostly, CAN is strongly associated with risk of arrhythmias, major cardiovascular events, myocardial dysfunction and cardiovascular mortality, as demonstrated in the ADVANCE, VADT and ACCORD studies [75].

#### 3.1.4. Innate Immune System 

The innate immune system plays an important role in CVD as protector or enhancer of disorders. Among the most prevalent leukocytes are the primary effectors, neutrophils. Their number and activation, based on upregulation of membrane markers, in blood are increased in obesity [76]. These cells are highly infiltrated in adipose tissue from obesity mice models that have been fed a high fat diet [77]. However, in T2DM, neutrophils increase their production of extracellular traps (NETs) by releasing decondensed chromatin and cytotoxic proteins. Overproduction of NETs can impair wound healing and increase chronic inflammation [78]. Obese subjects also have a pro-inflammatory macrophages profile, phagocytes of the immune system, in adipose tissue [79]. In addition, the phenotype of their precursors, monocytes, is modified in obese and diabetic patients [80]. A higher pro-inflammatory profile is associated with monocyte insulin resistance which is enhanced by ageing in a diabetes-independent manner [81].

### 3.2. Cardiometabolic Changes

#### 3.2.1. Endothelial Metabolic Dysfunction

In physiological conditions, the main energy substrate of endothelial cells is glucose due to its low mitochondrial content. However, hyperglycaemia increases oxidative stress, which implies the impairment of mitochondrial oxidative phosphorylation [82]. Diabetes enhances PKC activity, production of nuclear factor kappa-light-chain-enhancer of activated B cells (NF-κΒ) and generation of oxygen-derived free radicals in vascular smooth muscle, whose dysregulated function is exacerbated by impaired sympathetic nervous system function [83]. The senescence induced by ageing and diabetes can disrupt the endothelial function because of reactive oxygen species (ROS), inflammatory mediators or inducible nitric oxide (iNO) [4]. The dysfunctional metabolism of endothelial cells produces advanced glycation end products (AGEs) that through their receptors (RAGEs) activate mitogen-activated protein kinase (MAPK) and NFκB cascades and increase the production of ROS [84], inflammatory (vascular cell adhesion molecule-1 (VCAM-1), intercellular adhesion molecule-1 (ICAM-1) and monocyte chemoattractant protein-1 (MCP-1)) [85], profibrotic (matrix metalloproteinase (MMP)-2 protein) and prothrombotic factors (von Willebrand, plasminogen activator inhibitor-1 (PAI-1) and tissue factor (TF)) [86]. They are contributors to arterial stiffness, vascular calcification, and plaque accumulation in atherosclerosis-prone vessels and, consequently, diabetes mellitus-related vascular complications [87]. High circulating AGEs were associated with major adverse cardiovascular events (MACE) in patients with T2DM [88]. The soluble receptor (sRAGE) in blood might act as a decoy molecule and reduce the activity of AGEs. However, some studies have demonstrated a decrease in sRAGE in obesity, diabetes and ageing [89].

#### 3.2.2. Cardiomyocytes Metabolic Dysfunction

The main energy sources of the heart are fatty acids and glucose, contributing approximately 40–60% and 20–40%, respectively, of overall cardiac adenosine triphosphate (ATP) production, which depends on cardiac output. Energy efficiency is measured by the ratio phosphocreatine (PCr)/ATP [90] and ATP delivery through creatine kinase. In obesity, the PCr/ATP is low and the ATP delivery is elevated [91]. Although it suggests an energetic inefficiency because high ATP is needed for some stroke work [92]. In obesity, the main energy substrate is fatty acids, resulting in reduced myocardial efficiency (cardiac work per myocardial oxygen consumption). Similarly, in diabetes, there is a very important reduction in glucose oxidation [93] and an increase in fatty acids supply. However, the harder work of mitochondria might reduce its functionality and enhance the fatty acid accumulation with a cardiac lipotoxicity effect [61,94]. Glucose is taken up by glucose transporter 4 (GLUT4) insulin-dependent or glucose transporter 1 (GLUT1). The reduction of glucose metabolism derives from high glucose plasma levels and AGE formation through non-enzymatic binding between sugars and protein/lipid amine residues. Binding with RAGE can promote fibrosis through transforming the growth factor β1/SMAD pathway [95] and, consequently, myocardium remodeling. Higher circulating AGE and sRAGE were predictors of mortality and/or HF readmission in acute HF patients [96]. Their cross-linking of extracellular matrix proteins and effects on calcium re-uptake might also explain their deleterious effects on HF. In patients with T2DM, elevated AGE was associated with diastolic and systolic dysfunction [97].

#### 3.2.3. Neurons and Metabolic Dysfunction

Glucose is the main energy substrate in neurons or astrocytes. However, this is metabolized through oxidative phosphorylation to produce ATP or through glycolysis to produce lactate or pyruvate in astrocytes [98]. One of the main glucose transporters is GLUT3, which is reduced in diabetic patients [99]. Therefore, energy supply and hypothalamic glucose sensing is impaired in these patients. Injury of hypothalamus, enhanced by obesity, disrupts the effects of several metabolic hormones (ghrelin, insulin, leptin, glucagon-like peptide 1 (GLP-1)), which are key regulators of appetite [100]. On the other hand, parasympathetic and sympathetic nerves regulate glucose and energy metabolism. For instance, afferent vagal nerves transmit signals from the hepatoportal system and regulate glucose metabolism, and efferent vagal nerves regulate systemic glucose homeostasis by enhancing glucose-stimulated insulin secretion from β cells or by activating glycogen synthesis in the liver. By contrast, efferent sympathetic nerves inhibit insulin secretion from β cells, promote glucose production from the liver and promote lipolysis in white adipose tissue [101]. The accumulation of AGEs is also related to atrial remodeling and, consequently, AF [102].

#### 3.2.4. Innate Immune System Metabolism

Glucose is the main energy substrate of neutrophils and monocytes through oxidative phosphorylation, aerobic or anaerobic glycolysis [103]. After pro-inflammatory stimulus, there is an increase in anaerobic glycolysis with an increase in lactate. Neutrophils can also use glutamine to produce glutamate, aspartate, lactate and CO_2_. However, in diabetes, the glycolysis and glutamine metabolism are decreased and lead to their apoptosis [104].

### 3.3. Cardiac Endocrine Changes

#### 3.3.1. Endothelium-Endocrine Activity

Endothelial cells can release vasodilator mediators (nitric oxide (NO), prostacyclin, etc.), vasoconstricting (free radicals, endothelin, etc.), growth (colony stimulating factor, etc.) and procoagulant factors (von Willebrand factor, platelet activator factor, etc.) [105]. Endothelium relaxation is mediated by NO. The impaired production or degradation of this vasodilator mediator is affected by oxygen-derived free radicals or AGEs [38], produced by high glucose levels. Additionally, their receptor activation can also produce free radicals [106]. In diabetes, endothelial cell dysfunction is characterized by lower NO and higher prostanoid and endothelin production [107]. This protein promotes inflammation and vascular smooth muscle cell contraction and growth [108]. Some cytokines are also secreted by diabetic vascular endothelial cells and have an important role in collagen breakdown through matrix metalloproteinases production. This triggers vascular thrombosis [14] and CVD [109].

#### 3.3.2. Myocardium-Endocrine Activity

The natriuretic peptides include: atrial natriuretic peptide (ANP), brain (or B-type) natriuretic peptide (BNP) and C-type natriuretic peptide (CNP). BNP and ANP are cardiac hormones that increase intracellular cyclic guanosine monophosphate (cGMP) in target tissues through natriuretic peptide receptor-A (NPR-A or guanylyl cyclase-A) [110]. The synthesis of the NP precursors ventricular and atrial myocardium is modulated by volume expansion or/and pressure overload, end-diastolic wall stress and inflammation [111]. The natriuretic and diuretic effect of this peptide is mainly mediated by protein kinase G type II (PKG-II) present in the epithelial cells of nephrons that leads to a decrease in sodium reabsorption after inhibiting the amiloride-sensitive sodium apical channel and Na^+^/K^+^ adenosine triphosphatase pump. During HF, the production of NP is beneficial but its efficacy decreases with the disease in progression [112]. In obesity, BNP, NT-proBNP and MR-proANP levels are reduced in patients with and without HF [113] and weight loss increases their levels [114]. This may be caused by suppression of the bnp gene by circulating factors such as androgens [115] or neprilysin [116], generated by adipose tissue. It was demonstrated that the glucagon-like peptide 1 (GLP1) receptor agonist, liraglutide, was able to cause a significant rise in ANP secretion in mice because of the presence of GLP1 receptors on right atrial cardiomyocytes [117]. Evidence from epidemiological studies demonstrated an inverse association between systemic NP levels (both ANP and BNP) and body weight [118]. Variations in regional and, especially, visceral adiposity were related to circulating N-terminal proBNP. This could be partly moderated by the hyperinsulinaemic state observed in visceral adiposity, as high insulin levels have been demonstrated to suppress NP secretion and activity [119]. The Dallas Heart Study recently showed that both BNP and N-terminal proBNP are inversely related to visceral and liver fat, while being positively associated with gluteofemoral body fat, independent of insulin sensitivity [120].

### 3.4. Cardiac Adiposity

Physiological epicardial fat is an energy storage/energy supply to the myocardium or source of anti-inflammatory adipokines [121]. In 1986, an anatomic study showed that body weight is associated with higher adipocyte size, epicardial fat accumulation and its infiltration into the right ventricle [122]. This fat increment was also related to insulin resistance in obese patients [123], and its angiotensinogen production during cardiac surgery can induce postoperative insulin resistance [124]. Epicardial fat is also associated with cardiac structures, left and right atrial dimension, mitral and tricuspidal E/A ratio in a BMI, age and sex-independent manner [125]. It precedes left ventricular overload and hypertrophy [126]. Ageing and obesity are two associated factors with an impaired differentiation of adipocyte progenitors that contributes to insulin resistance [127,128]. There is a strong relationship between glucose metabolic disorder and visceral adiposity [129]. Low adipose tissue renewal suggests a higher expansion of older adipocytes, its hypertrophy, insulin resistance, lipolysis and inflammation response [130]. In consequence, higher lipids, glucose and inflammation enhance cellular senescence and its low differentiation ability [131]. However, the modulation of its endocrine and deleterious activity in the cardiovascular system is also very important because the released free fatty acids and adipokines by this tissue can affect the heart and blood vessels [132]. High epicardial fat accumulation contributes to inflammation because it is associated with an increase of pro-inflammatory mediators, i.e., monocyte chemoattract protein 1 (MCP-1) or soluble IL-6/IL-6 levels [133]. The adipose tissue can also release proteins associated with cardiac remodeling [134]. One of these molecules is leptin, which regulates sympathetic activity or angiotensin II-dependent vasoconstriction [135] and increases BP, affecting ventricle hypertrophy. In fact, some authors have suggested its ability to induce cardiac remodeling independently of body weight [136]. Some of the described mechanisms are increased ROS and malondialdehyde levels, the calcium–calpain-dependent apoptosis or the inhibition of Na^+^/K^+^-ATPase, affecting myocardial fibroblast proliferation and cardiomyoblast apoptosis [137]. Other molecules which were differentially released by epicardial fat from patients with diabetes and CAD were apolipoprotein A-I or retinol binding protein 4 (RBP4) [138,139], indicators of cardiovascular events [140]. RBP4 can act through toll-like receptor 4 (TLR4) and activate the c-Jun N-terminal protein kinase (JNK) pathway, improving insulin resistance [141]. In patients with suspected CAD, the epicardial fat volume, measured by computerized tomography (CT), is related to obesity, metabolic syndrome [142] (both cardiovascular risk factors), ageing and waist/circumference value [143], and CAD burden [144]. Some studies have demonstrated that a low-calorie diet for 6 months [145] or aerobic exercise might reduce epicardial fat in subjects with substantial obesity [146]. Epicardial fat accumulation is a risk of HF with mildly reduced ejection fraction (HFmrEF) and HF preserved ejection fraction (HFpEF). In these patients, a high epicardial fat volume is also a predictor of death and/or hospitalization for HF [147]. Greater total EATs in HFpEF is associated with myocardial fibrosis markers but not in HFrEF [148]. Epicardial fat thickness, which is enhanced in patients with diabetes and HFpEF, is associated with biventricular hypertrophy [149]. The proteome of this fat pad in HFpEF shows proteins related to lipid metabolism, mitochondrial dysfunction and inflammation [150]. Patients with obesity or T2DM with HFpEF often have AF [151]. Thus, epicardial fat volume participates in AF development [152] since adipocyte infiltration can interrupt conduction among cardiomyocytes or secret proteins with effects on their contractility or myocardium remodeling [153].

### 3.5. Inflammatory Changes

Obesity and T2DM are associated with an increase of inflammatory markers. This phenotype is more associated with HFpEF [154]. The shift of cardiometabolic source provokes a high lipid metabolic rate, affecting ROS production and inflammasome activation [155]. As a result, there is an IL-1β and IL-18 production. These products start the cardiac structural remodeling [156] because of cell death [157]. Some drugs have emerged to block these molecules [158,159]. However, Anakinra in the D-HART (Diastolic Heart Failure Anakinra Response Trial) pilot and D-HART2 trials did not show benefits after 12 weeks of treatment. Other inflammatory molecules that participate in cell death are those produced by neutrophils through NET formation and release [78], which are involved in a pronounced infiltration of inflammatory cells. Thus, the myocardium from obese patients has high macrophages marker levels [160]. This is considered an inflammatory state [161]. Similarly, diabetes induces inflammatory cell infiltration into the myocardium (high pro-inflammatory and low anti-inflammatory macrophages) [162]. However, colchicine, which inhibited neutrophil migration and reduced myocardial stiffness and cardiac hypertrophy in an animal preclinical model [163], was tested on chronic HF patients without success in ventricular remodeling [164]. The adipose tissue also releases proinflammatory adipokines (TNF-α, IL-6, MCP-1, leptin and resistin) [165] and, in consequence, develops HF [166]. The systemic metabolic dysfunction in the early stages of T2DM leads to immune cell senescence, contributing to the worsening of cardiac function and tissue metabolism [167].

### 3.6. Electrophysiological Changes

Diabetes may also be associated with proarrhythmic electrophysiologic changes. Several animal studies have shown that diabetes is associated with higher interatrial conduction times, increased atrial effective refractory period dispersion and prolonged action potential duration, which are correlated with increased susceptibility to AF [168,169]. One possible explanation is the reduction of K^+^ currents and irregularities in Na^+^ and Ca^2+^ currents. The latter could be explained due to the lower activity of the phosphatidylinositol 3-kinase (PI3K)/Akt pathway [170]. Adiposity increases the inflammasome product, IL-1β, which can generate arrhythmia by reducing the repolarizing K^+^ current (Ito) and increasing calmodulin kinase (CaMKII) oxidation/phosphorylation and Ca^2+^ spark frequency [171].

## 4. Sex Differences Regarding Diabesity and Cardiovascular Disease

### 4.1. Structural Changes

The combine data of imaging techniques and blood biomarkers analysis can distinguish two phenotypes (D and D/S) of asymptomatic patients according their risk for HF. The phenotype D (diastolic changes) with lower e′ and higher E/e′ ratio is associated with higher inflammatory blood biomarkers (fatty acid-binding protein 4 (FABP-4), IL-6, and IL1RL2). The phenotype D/S (diastolic and structural changes) with highest left ventricular mass and volumes, highest left atrial volume and lowest (absolute) left ventricular systolic strain, and lower e′ and higher E/e′ ratio are associated with high levels of soluble ST-2 (ST-2), troponin-I and C-type natriuretic peptide (CNP). The first phenotype, D, is more often identified in women and phenotype D/S more often identified in men [172]. Increased body weight in a diabetic murine model has demonstrated a structural adaptation of the heart and vasculature. It differs between males and females. In males, there is a reduction in vessel wall thickness and collagen content in the aorta and coronary artery in comparison with females. Thus, the vessel relaxation after acetylcholine treatment is more impaired in females than in males [173]. A sexual dimorphism was also detected in patients regarding diabetes and HFpEF. Women have more concentric remodeling and hypertrophy than men. The global longitudinal strain (GLS), circumferential strain (GCS) and radial strain (GRS), measured by cardiac MRI, showed a more severe impairment in diabetic women than in men.

### 4.2. Adiposity

A mice model with high fat diet has demonstrated that weight gain is higher in males than in females because males have low metabolic flexibility and mitochondrial respiration within brown adipose tissue [174]. Moreover, the adipose tissue distribution differs between pre-menopausal women and men [175]. While men accumulate adipose tissue around viscera, women accumulate it in the subcutaneous region. The adiposity in the upper-body or abdominal region is more associated with vascular dysfunction [176] and cardiovascular risk [177]. Testosterone deficiency, in males, might play an important role in abdominal fat accumulation [178]. A genome-wide association study and Mendelian randomization association has demonstrated that visceral adipose tissue can be the cause of type 2 diabetes, being higher in women than in men [179]. However, men are also susceptible to obesity and insulin resistance due to their low mitochondrial function [180]. Some authors have suggested that women will need a more aggressive and personalized medical treatment due to their high risk of cardiac failure [181]. In fact, the obese phenotype of HFpEF is more prevalent in women than men (2:1) [182]. Oestrogen deficiency might increase their risk [183].

## 5. Patient Management

### 5.1. Physical Activity

Patients with chronic diseases can decrease physical activity (PA) and adopt sedentary behavior [184] without the possibility of attenuating inflammatory profiles and iNOS protein contents, obtained by physical training [185]. Clinical studies have already demonstrated the cardiometabolic benefits of exercise: endothelium-dependent vasodilation, cardiac contractility, heart rate, blood pressure, blood flow, etc., [186]. Long-term exercise (>6 weeks) is able to upregulate glucose transporters and insulin receptors, resulting in a reduction in insulin resistance associated with T2DM [187]. Aerobic training can also reduce body weight [188]. Diet also has important beneficial effects on cardiac death or myocardial infarction, as was demonstrated the Mediterranean diet intervention (fruit and vegetables) [189]. PA and fitness might reduce diabetes and cardiovascular risk [190] and, in patients with myocardial infarction, modulate the cardiovascular risk factors, functional capacity and reduction in mortality [191]. In patients with HF, exercise training improves exercise tolerance, health-related quality of life [192] and all-cause hospitalizations [193]. These benefits were also observed in non-permanent patients with aerobic interval training for three months because it improves Vo2peak, ventricular and left atrial function, QOL and lipid levels [194].

### 5.2. Novel Hypoglycaemic Drugs

#### 5.2.1. SGLT2i

Sodium–glucose cotransporter 2 (SGLT2) is a membrane protein involved in the glucose and sodium active transport across epithelial cells (in kidney and intestine). Glucose is absorbed against the concentration gradient using energy added by the sodium gradient across the brush border membrane, maintained by Na^+^/K^+^ ATPase [195]. This transporter is responsible for the reabsorption of over 90% of filtered glucose at the glomerulus [196]. The main SGLT2 inhibitors (SGLT2i), empagliflozin, dapagliflozin and canagliflozin, have similar pharmacokinetic characteristics: long-elimination half-life, fast oral absorption, broad hepatic metabolism, low renal elimination and nonrelevant drug interactions [197]. They have a natriuretic effect, which is in part osmotic (corresponding to plasma glucose levels) and depends on the inhibition of sodium’s reabsorption at the proximal tubule, where sodium and glucose are co-transported in a 1:1 ratio [195]. The EMPA-REG OUTCOME trial (Empagliflozin-Cardiovascular Outcome Event Trial in Type 2 Diabetes Mellitus Patients) showed that empagliflozin, SGLT2i, reduced cardiovascular death by 38%, rates of major adverse cardiovascular events by 14%, all-cause mortality by 32% and hospitalization for HF by 35% in people with T2DM and established atherosclerotic CVD [198]. Canagliflozin and dapagliflozin have also demonstrated cardiovascular benefits, especially on HF and/or cardiovascular death [199,200]. SGLT2i has been proved to reduce congestion without worsening renal function in acute decompensated HF with reduced ejection fraction (HFrEF) in the 24 h after administration [199]. This drug produces a mild but important reduction in BP and extracellular fluid volume, which is observed at the beginning of treatment. It is presumed to be the reason for the initial 5 mmHg decrease in systolic BP (2 mmHg decrease in diastolic BP) observed within the first 14 days of therapy [197,201]. Despite these effects on BP, there is no evidence of heart rate increase. Indeed, some data suggest a drop in sympathetic activity [202]. Studies using bioimpedance spectroscopy confirmed that the reduction in BMI with SGLT2i therapy is caused by a decrease in adipose tissue mass and preserved lean tissue mass. The transient loss of extracellular fluid was normalized by six months [203]. Most imaging studies of SGLT2i use in humans have not shown a significant change in LVEF or volume, but showed an improvement in diastolic function [204]. The EMPA-HEART CardioLink-6 study demonstrated that the addition of empagliflozin to standard antidiabetic treatment in people with T2DM and CAD was associated with a significant reduction in left ventricular mass index (LVMi) as measured by cardiac magnetic resonance (cMRI) [205]. SGLT2i is associated with decreased production of leptin and reduced perivascular, perivisceral and pericardial adipose tissue deposition [206]. This result might explain their benefits for different disorders since EAT is associated with the severity of CAD, the risk of cardiometabolic disease and the development of AF [207]. Some experimental data showed a higher metabolic efficiency [208], lower lactate production [209] and reduced inflammation [210], focused on inflammasome products [211], monocyte chemoattractant proteins or endothelial proinflammatory proteins (endothelin) [212].

Although SGLT2i improves glucose metabolism, there are no data regarding food intake [213].

#### 5.2.2. Incretins

The main incretins, glucagon-like peptide-1 (GLP-1) and glucose-dependent insulinotropic polypeptide (GIP), are produced by the intestines 15–30 min after feeding for stimulating insulin secretion [214]. They are only active (1–2 min) until its inhibition by enzyme dipeptidylpeptidase-4 (DPP-4). While the GLP-1 receptors are found in the pancreas (a and b cells), heart, stomach, adipose tissue, vagus nerve and other regions of the central nervous system, GIP can only act on pancreas b cells [215]. However, its advantage lies in its protective role against hypoglycemia [216]. Both molecules are complementary because they reduce blood glucose without hypoglycaemia risk, promote weight loss and benefits in cardiovascular disease [217]. The clinical trial LEADER has demonstrated a cardiovascular benefit of liraglutide in patients with T2DM because it reduced 13% of nonfatal myocardial infarction or stroke and 22% of cardiovascular death [218]. There is evidence of the benefits of GLP-1 agonists for cardiac structure and function in animal and human studies [219,220,221]. Some authors have suggested its effect on body weight reduction as the main cause of diastolic function improvement by 20% of filling pressure reduction in T2DM [222]. However, in diabetic patients with subclinical systolic dysfunction, liraglutide improved the GLS in a weight and HbA1c independent-manner [223]. These results suggest other mechanisms of liraglutide effects on cardiac structure and function. More recently, a study on an animal preclinical HFpEF model has demonstrated that liraglutide, more than dapagliflozin, improves cardiac function and reduces cardiac hypertrophy, myocardial fibrosis, atrial weight, natriuretic peptide levels and lung congestion [224]. The cardiometabolic improvement by GLP-1 agonist was also tested in a preclinical model with lipotoxicity by palmitic acid. Thus, the GLP-1 agonist reduced lipid accumulation after reversing mitochondrial dysfunction [225]. The GLP-1 agonist is able to redistribute body fat and reduce cardiovascular risk [226]. Although a clinical study (MAGNA VICTORI) has demonstrated higher subcutaneous than epicardial fat reduction with GLP-1 agonist treatment [227], others have demonstrated a drop in epicardial and/or liver fat thickness in diabetic and obese patients after six months of treatment [228]. It suggests a direct mechanism of GLP-1 agonist in relation to epicardial fat since this tissue expresses receptor 1 [229], which is associated with fatty acid oxidation genes [230]. GLP-1 agonists can reduce inflammatory mediators [231] and monocyte adhesion to endothelial cells [232]. One of them is the CD11b expression levels of polymorphonuclear cells, which are modulated by GLP-1 in a myocardial infarction rat model. It suggests a vascular protection through the inflammation pathway [233]. The GLP-1 receptor expression of neutrophils and eosinophils from patients with asthma and diabetes might explain a ligand–receptor mechanism [234]. Another anti-inflammatory effect was associated with its ability to reduce lymphocyte proliferation (TH1 and 17), its glycolysis and its infiltration into injured organs [235].

New oral glucose-lowering drugs are a very attractive therapy for reducing the CVD progression focused on structural, endocrine, metabolic and inflammation benefits.

## 6. Conclusions

Ageing is associated with a higher ectopic adipose tissue accumulation that promotes insulin resistance. This metabolic dysfunction provokes elevated circulating AGEs that enhance ROS production and reduce NO in endothelial cells, favoring atherogenesis, and activate the fibrotic pathways of myocardium cells, favoring ventricle or atrial remodeling. The hypothalamic dysfunction might reduce anorexigenic and metabolic peptides, affecting ectopic epicardial fat accumulation and inflammation. Lifestyle (diet and physical activity) can modulate these mechanisms, although exercise intolerance or low physical capacity during ageing suggests a clear need for new therapeutic strategies with metabolic, structural and endocrine improvements (Figure 2).

## Figures and Tables

**Figure 1 ijms-23-07886-f001:**
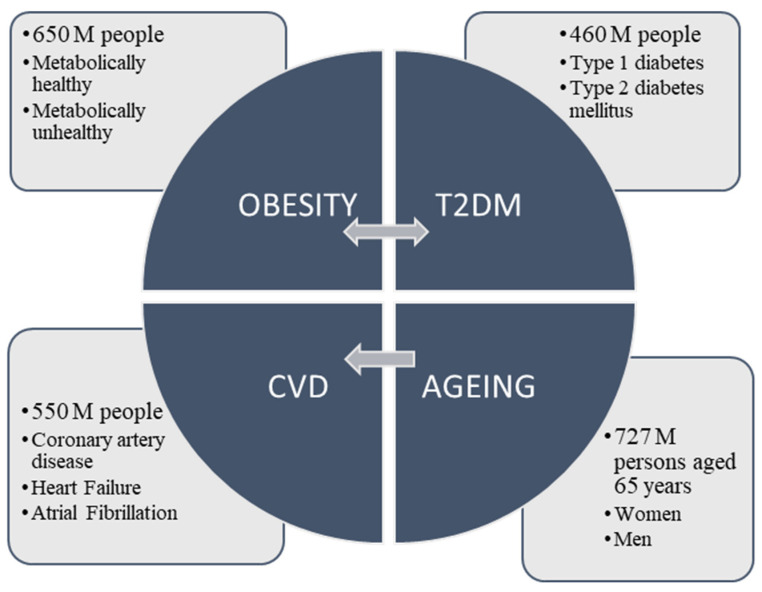
Prevalence of obesity, type 2 diabetes mellitus (T2DM), cardiovascular disease (CVD) and ageing.

**Figure 2 ijms-23-07886-f002:**
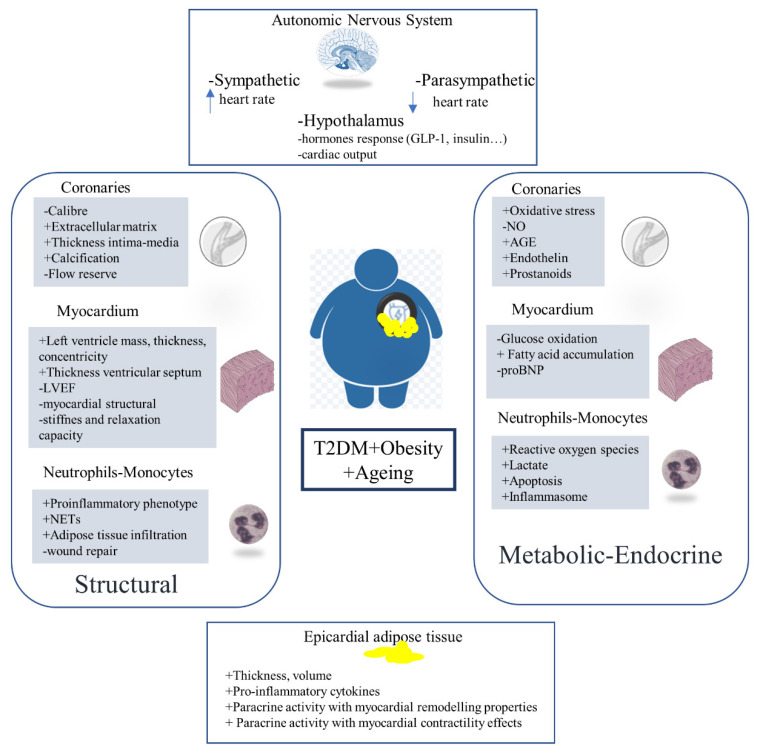
Structural, metabolic-endocrine and adiposity changes in T2DM, Obesity and ageing.

## Data Availability

Not applicable.

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
