# Peer review of "Diabesity in Elderly Cardiovascular Disease Patients: Mechanisms and Regulators"

_ijms, 2022, doi:10.3390/ijms23147886_

Round 1

Reviewer 1 Report

The authors' purpose is an interesting aspect of the impact of diabetes and obesity on cardiovascular diseases.

but:

- the title should be changed, because even with an interesting and exhaustive discussion the role of EAT is not pointed out, so probably is better to have a generic title

- physical activity should be mentioned, as having a fundamental impact on cholesterol, ROS, and consequently cardiac health; besides that hormonal milieu is strongly influenced too 

- the nutritional point of view should be included

- AGEs are correctly considered but the paragraph should be expanded

- the review lacks a "take-home message"

- the possible influence on diabesity of RBP could be cited too (for example 10.2174/1389557515666150709112822)

Author Response

Reviewer 1

- the title should be changed, because even with an interesting and exhaustive discussion the role of EAT is not pointed out, so probably is better to have a generic title

Response: We apologize the specific title of the manuscript. It was changed by “Diabesity in the elderly cardiovascular disease patients: mechanisms and regulators”

- physical activity should be mentioned, as having a fundamental impact on cholesterol, ROS, and consequently cardiac health; besides that hormonal milieu is strongly influenced too 

Response: We are sorry because lifestyle was not included in the previous version. It was changed according reviewers´ s suggestions and paragraph 5.1 and references were included into new manuscript.

- the nutritional point of view should be included

Response: Current version has included the nutritional point of view, specifically, Mediterranean diet (paragraph 5.1.)

- AGEs are correctly considered but the paragraph should be expanded

Response: AGE´s paragraph was expanded (lines-238-250),(lines-267-271) and (lines-286-287).

- the review lacks a "take-home message"

Response: At the end of the review, we have included a conclusion or take-home message “The ageing is associated with a higher ectopic adipose tissue accumulation that promotes insulin resistance. This metabolic dysfunction provokes an elevated circulating AGEs that enhance ROS production and reduces NO of endothelial cells, favoring atherogenesis, and activates fibrotic pathways of myocardium cells, favoring ventricle or atrial remodeling. The hypothalamic dysfunction might reduce the anorexigenic and metabolic peptides with consequences on ectopic epicardial fat accumulation and inflammation. Lifestyle (diet and physical activity) can modulate these mechanisms, although the exercise intolerance or low physical capacity during ageing suggests a clear need of new therapeutic strategies with metabolic, structural and endocrine improvements”.

  - the possible influence on diabesity of RBP could be cited too (for example 10.2174/1389557515666150709112822)

Response: This reference was included in the new version (RBP4 can act through Toll-like receptor 4 (TLR4) and activate c-Jun N-terminal protein ki-nase (JNK) pathway, improving the insulin resistance [141].

Reviewer 2 Report

The manuscript details the role of cardiovascular diseases particularly diabesity which is a contributing factor to various cardiac, metabolic and neurohormonal changes. It also highlights the role of epicardial adipose tissue in promoting cardiovascular diseases.

The author has provided a detailed information regarding diabesity and the pathophysiological mechanisms involved in the development and progression of cardiovascular diseases.

Although detailed information regarding pathophysiological mechanisms involved in diabesity is provided, however information regarding patient management is not clearly stated.

Also apart from novel drugs being used, the author should also include information detailing the role of day to day lifestyle changes in optimizing glucometabolic control of diabesity.

Author Response

The manuscript details the role of cardiovascular diseases particularly diabesity which is a contributing factor to various cardiac, metabolic and neurohormonal changes. It also highlights the role of epicardial adipose tissue in promoting cardiovascular diseases.

The author has provided a detailed information regarding diabesity and the pathophysiological mechanisms involved in the development and progression of cardiovascular diseases.

Although detailed information regarding pathophysiological mechanisms involved in diabesity is provided, however information regarding patient management is not clearly stated.

Response: We have included a paragraph according patient´s management after reviewer´s suggestions (paragraph 5).-

Also apart from novel drugs being used, the author should also include information detailing the role of day to day lifestyle changes in optimizing glucometabolic control of diabesity.

Response: We are sorry because lifestyle was not included in the previous version. Now, it was described in paragraph 5.

Round 2

Reviewer 1 Report

The authors have made enough improvement to make the manuscript published

Reviewer 2 Report

Authors have addressed my comments and made necessary changes. I recommend this review for acceptance.